# Ancillary services from wind turbines: AGC from a single Type 4 turbine

**Eldrich Rebello, David Watson, and Marianne Rodgers**

Wind Energy Institute of Canada, 21741 Route 12 Tignish, PEI C0B 2B0, Canada

**Correspondence:** Eldrich Rebello (eldrich.rebello at weican dot ca)

**Abstract.** Wind turbines possess the technical ability to provide various ancillary services to the electrical grid. Despite this, renewable generators such as wind and solar have traditionally not been allowed to provide significant amounts of ancillary services, in part due to the variable and uncertain nature of their electricity generation. Increasing levels of renewable generation, however, continue to displace existing synchronous generation and thus necessitate new sources of ancillary or system services. This work is part of an ongoing project that seeks to provide empirical evidence & an examination of how ancillary services can be provided from commercially available wind turbines. We focus specifically on providing secondary frequency response (AGC) and demonstrate that wind turbines have the technical capability to provide this service. The algorithms used are intentionally simple so as to evaluate the capabilities and limitations of the turbine technology. This work presents results from a single, 800 kW, IEC Type 4 wind turbine. 10% of rated power is offered on the regulation market. We do not separate up- and down-regulation into individual services. Up-regulation is offered through a 5% constant power curtailment. The AGC update interval is 4s, to mimic real-world conditions. We use performance scoring methods from the Pennsylvania-Jersey-Maryland (PJM) operator and the National Research Council (NRC) of Canada to quantify the wind turbine's response. We use the calculated performance scores, annual site wind data and 2017 PJM market price data to estimate income from providing secondary frequency regulation. In all cases presented, income from the regulation market is greater than the energy income lost due to curtailment.

## 1 Introduction

One means of slowing the pace of climate change is through the decarbonisation of the electric grid. Established means of generating electric power include driving large turbines via coal boilers, burning natural gas, nuclear fission or hydro generators. Decarbonisation refers to reducing the amount of greenhouse gas emissions from generating sources like coal and gas and generating increasing amounts of energy from non-emitting sources such as wind and solar. Several jurisdictions around the world have set targets of supplying an increasing share of their electrical energy from renewable sources such as solar PV and wind. Examples include the EU's targets of 20% by 2020 (Capros et al., 2011) and the Paris Climate Accord (Baruch-Mordo et al., 2018). Although renewable generating technology has matured over the past few decades, it does come with some limitations. Most of the concerns stem from the fact that renewable generation is inherently variable and uncertain and is increasingly displacing large, synchronous generating capacity on the electric grid. Continually displacing synchronous generation brings with it technical challenges such as falling grid inertia that is provided by the electromechanical properties of salient pole machines. Some of these challenges are documented in a 2013 paper from the IEA's Task 25 work (Holttinen et al., 2013) and also Piwko et al. (2012). Further, the system services or ancillary services once provided by conventional generators will now require alternate sources. The question of whether renewable generators can provide these services has been examined in detail in the past in examples such as Banshwar

et al. (2017) & Bevrani et al. (2010) and the answer is almost always yes.

Even given this situation, grid operators around the world have been hesitant to source ancillary services from renewable generators. Part of this can be explained by the variable and uncertain nature of renewable generators. Their outputs depend on factors such as wind speeds and solar irradiance and these can never be predicted with perfect accuracy. Other generators on the power system typically 'accommodate' renewable generation, changing their outputs to account for variations in renewable generation. Further, the competitive markets that are designed to source these services are often set up to skew heavily towards large, synchronous generators (Denholm et al., 2019). Qualification rules might, for example, include the requirement that generators be ready to provide system services when called upon to do so and sustain a response for a certain amount of time. As an example, consider system services that depend on active power, particularly increases in it. This is not a problem for fossil-fuel powered generators as their active power outputs are largely controllable by fuel flow. Such a requirement might, however, disqualify generators such as wind and solar whose active power outputs depend on uncontrollable sources: wind speeds and solar irradiance. These markets are typically not designed to value system services provided by generators whose fuel cost is zero. Finally, since the output of renewable generators depends on a variable input, their ability to participate in day-ahead markets is heavily dependent on accurate forecasts.

## 1.1   Grid frequency response & AGC

The operation of any electric grid is a balance between supply and demand. Grid frequency is often used as a good indicator of the relative balance between supply and demand. If supply exceeds demand, frequency rises. Conversely, if demand exceeds supply, grid frequency falls. Grid frequency is typically controlled in a narrow range. The time response of grid frequency to a disturbance (e.g. sudden increase in load, loss of generation) is shown in Figure 1. Depending on the type of technology used, wind generators possess the ability to participate in the primary, secondary and tertiary response regions of Figure 1. This work focuses specifically on the region of secondary frequency response. At its core, this is an attempt by the grid operator to balance supply and demand through small changes in the power outputs of several generators. A grid operator calculates the difference between anticipated power production and load values, compares these to their measured values and accounts for power flows into other areas. This forms the basis of what is called an Area Control Error (ACE) from which a secondary frequency regulation signal (AGC signal) can be derived. This AGC signal is then scaled and sent to selected generators to regulate their active power outputs accordingly. Typically, generators

providing services such as AGC are large, synchronous machines.

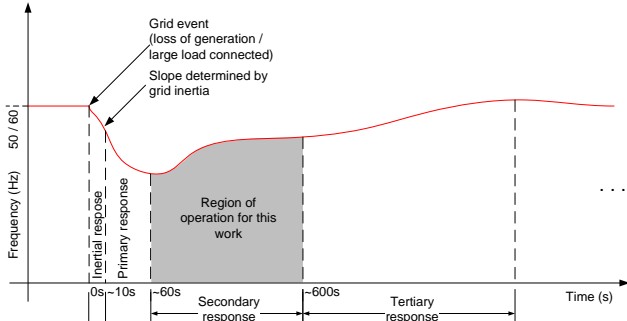

**Figure 1.** Grid frequency response to a sudden load increase or loss of generation. Region of secondary frequency regulation shown shaded.

## 1.2   Similar work

Providing ancillary services from wind turbines is neither new nor novel. Much of the publicly available literature, however, consists of simulations (e.g. Aho et al. (2015), Shapiro et al. (2016), Basit et al. (2014)) that demonstrate the theoretical ability of wind generators to provide secondary frequency regulation or which evaluate its effects at a system-level. Empirical performance data evaluating the performance of wind farms, particularly of commercially available technology, in providing ancillary services is often limited or not publicly available. The data presented in this work is therefore of importance to system operators as it an unbiased evaluation of commercial wind turbine technology's performance. In contrast, wind turbine manufacturers have limited motivation to provide this data as they do not operate the power system and, more importantly, there is an inherent conflict of interest in performance numbers published by them.

In contrast to results from NREL's CART-3 wind turbine Aho et al. (2012) which is a prototype demonstration turbine, our work uses commercially available wind turbine technology and evaluates time-series response data at a granular level. Further, our work contributes by disclosing the exact algorithms used (simple as they may be) and presenting a technical and financial analysis of the results of frequency regulation tests. One example of similar work using commercial wind turbine technology is the EU's TWENTIES project from 2013 (Azpiri et al., 2014). The TWENTIES project examined the ability of three geographically separated, transmission connected wind farms on the Spanish grid to control their active power in response to an external AGC signal. The results of the 30 minute test were encouraging however a detailed analysis using metrics such as performance scores was not presented. Similar to our work, the TWENTIES demonstration provided up-regulation via curtailment and also used

a 4s AGC update interval. On a larger scale, the US utility Xcel Energy continues to provide AGC from wind farms on their network (Lew et al., 2013) however a critical analysis of the results is not publicly available. The Wind Energy Institute of Canada[1] has published results (Nasrolahpour et al. (2017) & Rebello et al. (2019)) of AGC tests on their own wind farm that consists of five IEC Type 5 wind turbines (directly connected synchronous machines). These works follow a similar analysis method as presented here however the key differences with this work are the turbine technology (Type 5 versus Type 4) and the number of turbines (5 versus 1).

Solar PV represents an inverter-based technology that is similar to what is used in a Type 4 wind turbine (variable speed, full converter). First Solar conducted a demonstration of a solar farm's ability to provide secondary frequency regulation (AGC) (Loutan et al., 2017) in collaboration with the California System Operator and NREL. First Solar's demonstration shares a lot in common with our work. Both use curtailment to provide room for up-regulation and both use inverter-based technologies. Although the internal control algorithms are different (completely electronic for a solar inverter versus some mechanical control for a wind turbine), the results of First Solar's work and this work are encouraging as the end result for the grid operator is the same: wind and solar generators both possess the technical ability to provide system services such as AGC. It is worth mentioning that synchronous areas such as the EirGrid network (in Ireland and Northern Ireland) and ERCOT (Texas, USA) regularly operate their systems with significant levels of generation coming from renewable sources.

## 1.3 Limitations

This work is limited to examining the abilities of existing wind generators in providing one specific ancillary service: secondary frequency regulation. Our primary aim is to make public a granular (second-by-second) analysis of the performance of wind generators when providing secondary frequency regulation. Although we aim to present a broad technical and financial analysis, there are numerous considerations that are not examined here. Examples include wake effects in wind farms (See van Wingerden et al. (2017)) and the effects of market prices (See Holttinen et al. (2016). Jansen (2016)). The effect of providing system services on the lifetime of wind turbines is also not examined here. This is of particular interest to wind turbine manufacturers and equipment owners. For example, derated operation to provide secondary frequency regulation (as presented here) involves increased pitch system action and changes to structural loads both of which could involve maintenance costs. Further, although our work presents results for a longer time duration than previously published work, this is still insufficient for

a complete analysis. Aspects such as seasonality (e.g. summer wind speeds being different from winter wind speeds), temperature & maintenance requirements are not examined. Examining these will likely require a study spanning several months.

Performing research work on commercial wind turbine technology is difficult and as such, data in this work is limited to a single wind turbine. Aggregating the stochastic effects of several wind turbines across a wind farm is likely to produce different results (See (Rebello et al., 2019) for data from a five-turbine wind farm). Finally, this work is limited to data from an IEC Type 4 wind turbine.

## 1.4 Motivation & utility of results

Although grid codes of several countries/regions require new wind generators to be capable of numerous ancillary services, empirical, unbiased data on the abilities of wind generators is lacking. Often, provisions are laid down in grid codes or connection agreements but wind generators are rarely called upon to provide an ancillary or system services. An example of public information of this nature is Hydro-Quebec's comparison of fast-frequency response from two wind turbine technologies (Asmine et al., 2017). In a similar vein, our intent is to make operational data public to allow for greater scrutiny by system / grid operators and to give grid operators an unbiased method of comparison between turbine technologies. The work resulting from this project seeks to fill these gaps in a transparent and critical way. For system operators, this project (i.e. the work beyond what is presented here) seeks to evaluate the performance of various wind turbine technologies in providing the specific ancillary service of AGC. As an example, the response of an IEC Type 3 wind turbine will be different from that of a Type 4 turbine. Although both designs control active power via pitch regulation, their electrical connection to the power grid differs and it is therefore important to study and quantify this difference in response.

Our algorithms and control methods are designed to be as high-level as possible. Note that this work does not develop a method for controlling the active power output of a wind turbine (unlike Aho et al. (2012), for example). The specifics of this control (blade pitch angles, inverter phase angles, etc.) are left to the manufacturer's design. We focus solely on evaluating the end result and not on the specifics of the control method. This is the exact same viewpoint for a grid operator in the sense that *how* a generator's active power is controlled is not as important as the fact that active power *can be* controlled.

## 1.5 Ancillary service markets procuring services from wind generators

Broadly, ancillary services (or system services) refer to a set of services that complement the primary grid purpose of sup-

---

[1]same authors as this work

plying energy. Examples include system inertia, voltage control, primary frequency control, operating reserves etc. Ancillary services may or may not depend on active power. Examples that do not depend on active power include reactive
power and voltage support while services such as operating reserves and regulation depend on active power. The question of how increasing renewable generation will affect markets for ancillary services has been considered in the past Ela et al. (2012). (Banshwar et al., 2017) present a good overview
of the challenges to sourcing ancillary services from renewable sources.

As detailed in Bloom et al. (2017), markets such as California's ISO (CAISO) and EirGrid in Ireland operate with >20% of annual energy coming from renewable generation
such as wind and solar. These generators operate with a nearly zero marginal cost of energy and are expected to influence both energy and ancillary service markets. In markets such as CAISO, day-ahead energy prices routinely reach zero. The challenge faced by grid operators is one of integrat-
ing the suite of services offered by wind and solar generators and designing markets to allow their effective participation while maintaining system reliability.

### 1.6 Test site location and description

The tests described in this work were performed at the
25 Saskatchewan Research Council's Cowessess First Nations site in Saskatchewan, Canada (Fig. 2). The site consists of a single wind turbine and a battery storage system and has been in operation since April 2013. The 800 kW wind turbine has a hub height of 73 m and a rotor diameter of 53 m.
See Jansen et al. (2013) for more details on the test site and equipment. The battery is not part of the results in this work.

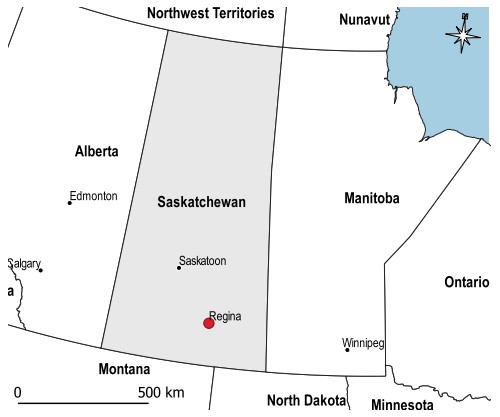

**Figure 2.** Test site location (red) in Canada. Province of Saskatchewan shown shaded.

A single-line diagram of the Cowessess site is shown in Figure 3. The wind turbine and battery inverter each connect to the 25 kV bus via their own transformers. Note the
35 location and connections of our controller. We control only the active power setpoint (or target) of the wind turbine. All

other control such as pitch, power error, etc. are left to the turbine's internal controller. Communication to the turbine's controller is via Modbus.

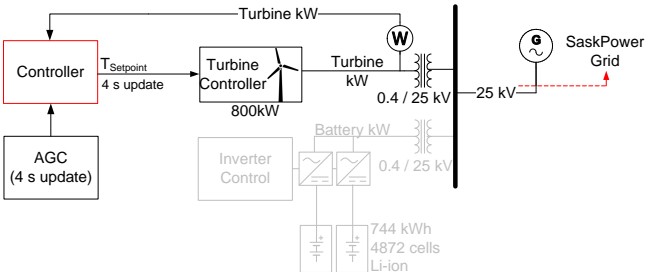

**Figure 3.** Single-line diagram of the Cowessess First Nations site in Regina, Saskatchewan. Note that the battery is not part of the results presented here.

## 2 Algorithm & AGC signal

The Alberta Electric System Operator (AESO) provided both AGC signals used in this work. One was a 30 minute duration signal and the other was 4.5 hours long. These are identical to the signals used in Rebello et al. (2019) and Nasrolahpour et al. (2017). This is done to make direct comparison with
earlier work easier. Both signals use a 4 s update interval which is identical to PJM's Reg-D signal. Although we do not use PJM's regulation signals, the identical update intervals allows for a more straightforward comparison. The first step to signal preparation was scaling the raw AGC signals
to fit within our chosen regulation range of 40 kW. We use the PJM definition here where ±40 kW corresponds to a regulation range of 40 kW. The results of this scaling are shown by blue traces in Figure 5 (a) and (b). Note that the signal in Figure 5 (a) has a range from 720 to 800 kW (centred around
760 kW i.e. 800 − 40 kW) as these power targets are sent directly to the wind turbine. Power values in the range [720, 800] kW are within the operational range of the wind turbine and this test is performed when prevailing wind speeds are above the turbine's rated wind speed i.e. rates power produc-
tion is possible. The signal in Figure 5 (b) is centred around zero kW as this signal is a bias value. The bias values are therefore in the range of [-40, 40] kW and are added to an estimated power value as described in Section 2.3. The scaling process was followed by filtering, as described below.

### 2.1 AGC signal filtering

A wind turbine is a system with electrical and mechanical components and therefore has a finite response time and stochastic variations in power output. To account for this, we apply a simple differential magnitude filter to the raw AGC
signals from the AESO. The purpose of this filter is to prevent repeated, small changes to the power target of the wind

turbine. Note that the power output of any wind turbine has small variations in it, i.e. it is very rarely a steady value. The changes in the AGC signal must be greater than the magnitude of these changes in order for the wind turbine to re-
⁵ spond in a meaningful way. From empirical data we calculate a standard deviation of 11 kW in the wind turbine's power output. This data consisted of one year of 1 s interval power data. We examined instances of operation at rated power as the turbine's control behaviour is similar to when power is
¹⁰ curtailed. In other words, above rated wind speeds, the turbine's pitch system actively limits power to a defined target and the standard deviation of this data is a good measure of the inherent variance in power output. The value calculated here is valid only for this particular turbine and location.

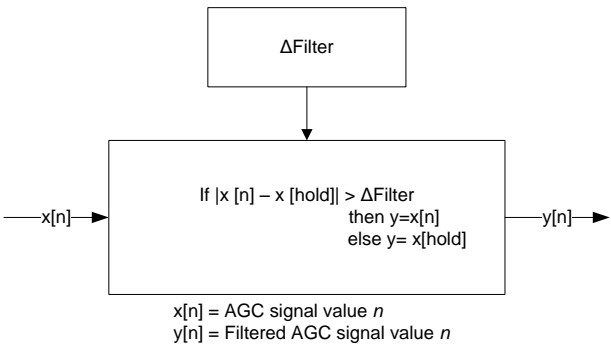

**Figure 4.** Differential magnitude filter logic

¹⁵   The differential magnitude filter output will not change until the difference between the present and next AGC values is more than 11 kW. Additional information about this method is in Rebello et al. (2018). The results of this filtering are shown with blue traces in Figure 5. Applying this filtering af-
²⁰ fects the calculated performance scores. This is because the grid operator sees only the unfiltered signal and the generator's response and calculates a performance score based on these two signals. Note also that the signals shown in Figure 5 represent a somewhat extreme case with frequent changes,
²⁵ almost every 4 seconds. Signals closer to the filtered (blue) ones in Figure 5 are more common.
  The performance score numbers in this work are calculated relative to the original AGC signal i.e. before filtering and represent the performance scores calculated by the util-
³⁰ ity. Note that the effect of the applied AGC signal filtering on the calculated performance score is not always consistent i.e. it can either increase or decrease the performance score. This is explained due to stochastic variations in measured power output from the wind turbine sometimes aligning better with
³⁵ the unfiltered AGC signal as opposed to the filtered AGC signal.

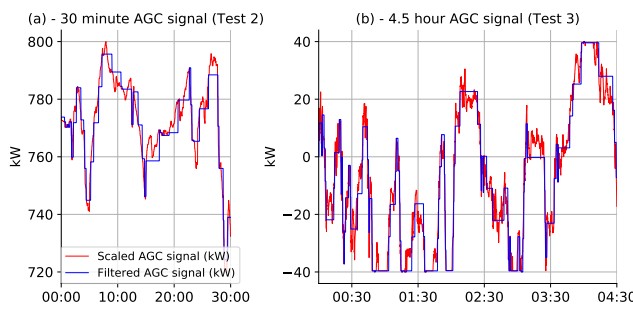

**Figure 5.** Scaled AGC signals and result of applying a differential magnitude filter.

## 2.2   Test outlines

The experiments presented in this work are grouped into two tests as summarized in Table 1 with two being above rated wind speed and one below. The aim of both tests is ⁴⁰ to examine the ability of the wind turbine to vary its active power output in response to an external target i.e. the applied AGC signal. In order to provide a complete picture, examining this ability across the full range of operational wind speeds is required. Test 3 is performed below rated wind ⁴⁵ speed and therefore requires a varying power curtailment to provide up-regulation. As described in Section 2.3, this varying power curtailment is provided via a wind speed estimate and a power curve. In contrast, Test 2 is performed when prevailing wind speeds are above the turbine's rated wind speed ⁵⁰ and rated power production is therefore possible. No estimate of power production is required. We also present a variation of Test 2 where the regulation offer is 100 kW. This is denoted by Test 2* as it is functionally identical to Test 2, the only difference being a larger regulation region (100 kW ver- ⁵⁵ sus 40 kW in Test 2). Test numbers are kept consistent with other project documentation. The two tests presented here are the only ones with the wind turbine operating independently.

**Table 1.** Summary of the tests presented in this work

|        | Duration | Description | Regulation offer |
|--------|----------|-------------|------------------|
| Test 2 | 30 m | Wind turbine operating above rated wind speed | 40 kW |
| Test 2* | 30 m | Wind turbine operating above rated wind speed | 100 kW |
| Test 3 | 4.5 h | Wind turbine operating below rated wind speed | 40 kW |

## 2.3   Algorithms & Methodology

Figure 6 illustrates the algorithms used in the two tests pre- ⁶⁰ sented here. Test 2 simply sends power setpoint targets to the

wind turbine. All AGC target values range from 720 kW to 800 kW. These are the filtered AGC values from Figure 5 (a). We use a wind forecast to select a time when the wind speeds are above the turbine's rated wind speed. As such, the turbine is expected to generate rated power and no power estimation is required.

The algorithm for Test 3 is slightly more complex as it is performed below the wind turbine's rated wind speed. The turbine's power output therefore varies with the wind speed. The challenge here is estimating the turbine's power generation potential. We use averaged (30 s) wind speed data from the turbine nacelle anemometer as an estimate of short-term wind speed trends. We then estimate the expected power generation from the turbine using the power curve. Note that this power curve was constructed from measured, historical data (1 year) and is not the manufacturer's power curve. We then subtract 40 kW from the expected power production value to provide room for up-regulation. Finally, we add the AGC signal bias value to calculate the power target for the wind turbine. This value is updated every 4 seconds and is sent to the turbine's control system. Although the wind turbine used here provides a $P_{available}$ signal that is an estimate of the power in the wind, we observed delays and errors in this signal and so elected to use a measured power curve and wind speed average as described above.

In both Test 2 & 3, our controller (red block in Figure 3) sends a power target to the turbine controller. This setup is similar to what a grid operator would use. Although the power target calculations in the two tests appear different, they are not. The algorithm for Test 3 would produce the same power targets as the signal in Figure 5 (a) by estimating the available power as 800 kW, subtracting 40 kW and then adding the AGC value.

## 3   Results: Performance scores, discussion & analysis

### 3.1   Performance scores

A performance score is our chosen metric for analysing the ability of a wind turbine to provide AGC. A performance score is a numerical measure of a generator's ability to follow an external control signal. We use two methods of calculating performance scores:

1. Natural Resources Canada method (Kabiri and Song, 2018)

2. PJM method (Pilong, 2015)

The utility of the PJM performance scores is detailed in Section 4. A summary of the results from the NRC and PJM performance score calculations is shown in Figure 7. Readers are directed to the references above for full details of the calculation methodologies. One major difference between the two methods is that the NRC performance scores are based entirely on the error between the target and the measured power while the PJM method accounts for delay, accuracy and precision. Comparing scores between the two methods is therefore not possible.

### 3.2   Test 2 - 40 kW regulation offer

Results from one instance of Test 2 are in Figure 8(a). Although the test was repeated several times, only one example is shown here. Observe that a drop in the wind speed caused a drop in the power output. This has a noticeable effect on the performance score. These results are also negatively affected by a scaling error in the turbine's control system where power setpoints were incorrectly scaled assuming a rated power of 840 kW. This is the reason why the measured turbine power is consistently greater than the target values and why a gap between the two is clearly visible. Despite this, the general trend of the red and blue traces agrees well. Due to time and weather constraints, it was not possible to repeat this test during identical conditions after correcting for the scaling error however, a substitute is presented below.

### 3.3   Test 2* - 100 kW regulation offer

Results from this test are presented in Figure 8 (b). This test is not exactly the same as Test 2 above. A major difference is that the regulation offer here is 100 kW versus 40 kW above. Further, the scaling error with power setpoints from Test 2 was corrected. Both these fact combine to improve the performance scores. A further reason for the improved performance scores here is the fact that although the magnitude of the error remains comparable to earlier iterations, the error percentage is now smaller relative to the regulation bid resulting in an improved performance score. This is visible in Figure 7 (d) & (e). What is clear from Figure 8 (b) is that a Type 4 wind turbine is able to control its active power accurately and with a relatively consistent error magnitude.

The data presented in Figure 8 (b) represents a situation where the wind speeds were sufficient to allow rated power production but cold temperatures required curtailment to below rated power. We argue that the performance of the turbine in these conditions is identical to that at higher power levels as it is constrained by the turbine's control system (e.g. pitch action). Due to the significant gap (or headroom) between the power setpoints in Figure 8 (b) and the possible power, we also reduce the chance of a drop in wind speed affecting the performance score. This, of course, comes at the cost of reduced energy income but we argue that this presents a fair assessment of the wind turbine's abilities. The resulting performance score is comparable to PJM performance scores reported for hydro generators Croop (2017).

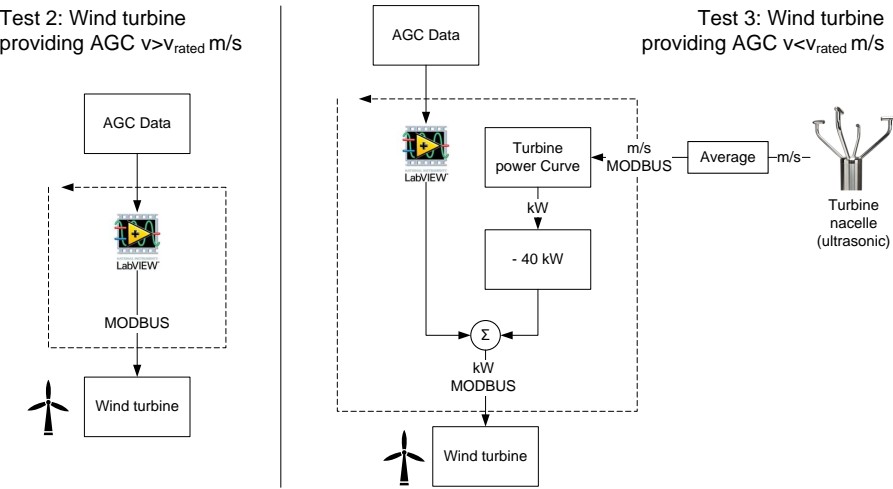

**Figure 6.** Algorithms for (a) Test 2 and (b) Test 3

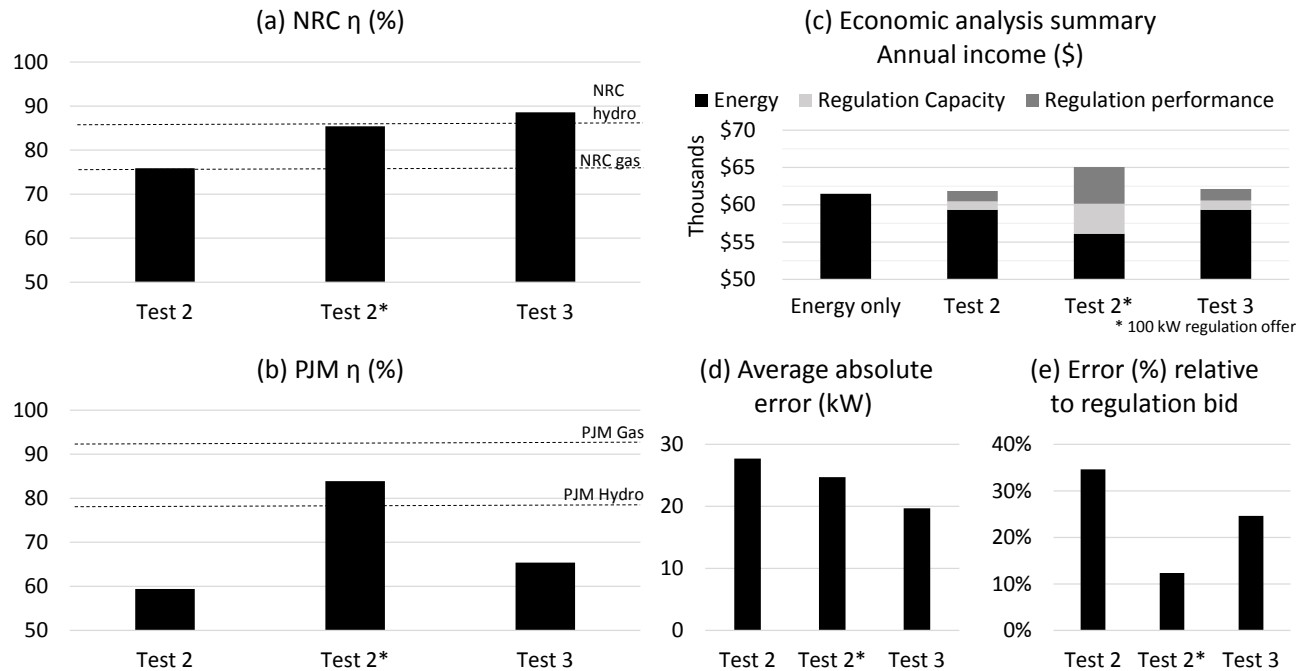

**Figure 7.** (a) & (b) Performance score summary. Dashed lines show performance figures for other generation technologies; (c) Annual economic summary with various performance scores; (d) Error trends in each test & (e) Error relative to regulation bid magnitude

## 3.4  Test 3

Test 3 represents turbine performance below rated power. The power setpoint scaling error from Test 2 was corrected here. Observe from Figure 9 that the target and measured power values track each other well. Correcting the scaling error improves the calculated performance scores relative to Test 2, however, note that the error magnitude relative to the regulation bid magnitude is similar to that in Test 2 (see Figure 7 (e)). This combined with the small regulation bid magnitude limits the performance score obtainable.

## 3.5  General Comments

Note from Figure 7 (d) that the general error trend across all three tests is broadly comparable. The effect of the scaling error is most pronounced in Test 2 as this was performed at rated power and the full magnitude of the error affected the results. Previous runs of Test 3 were affected by this error but the effect is less pronounced as the error is proportional to power. After correcting for this error, a repeat of Test 3 showed reduced error magnitude (Figure 7 (d)), however, the error magnitude was broadly comparable to Test 2*

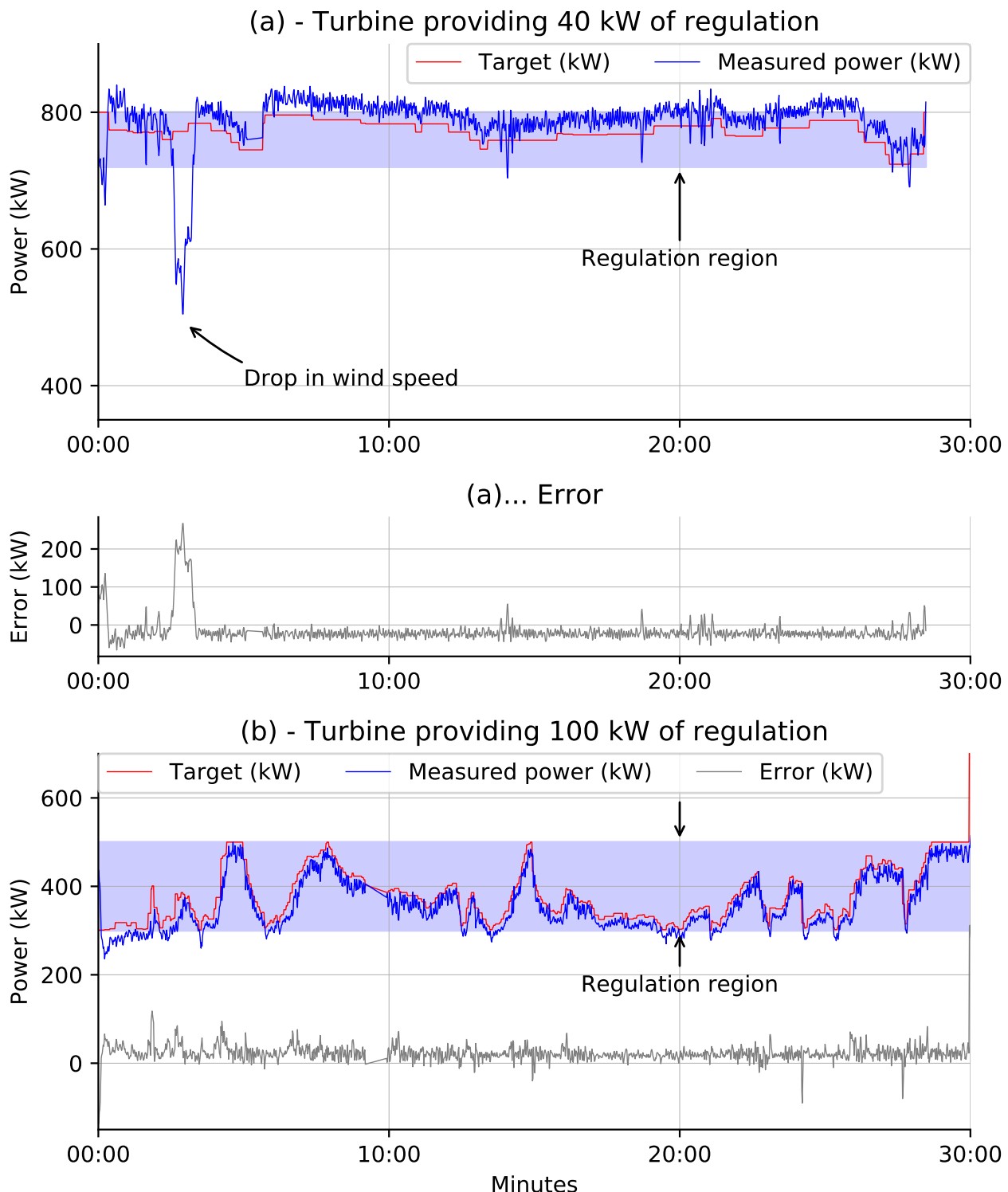

**Figure 8.** (a) Providing 40 kW of regulation from a single 800 kW wind turbine when operating at rated power. Note that the offset observed is discussed in Section 3.2. (b) Providing 100 kW of regulation. Blue shaded region in (a) & (b) is the range of possible regulation

(100 kW). This suggests that a major contributing factor to the performance score is the magnitude of error relative to the regulation bid. An error of 20 kW with a regulation bid

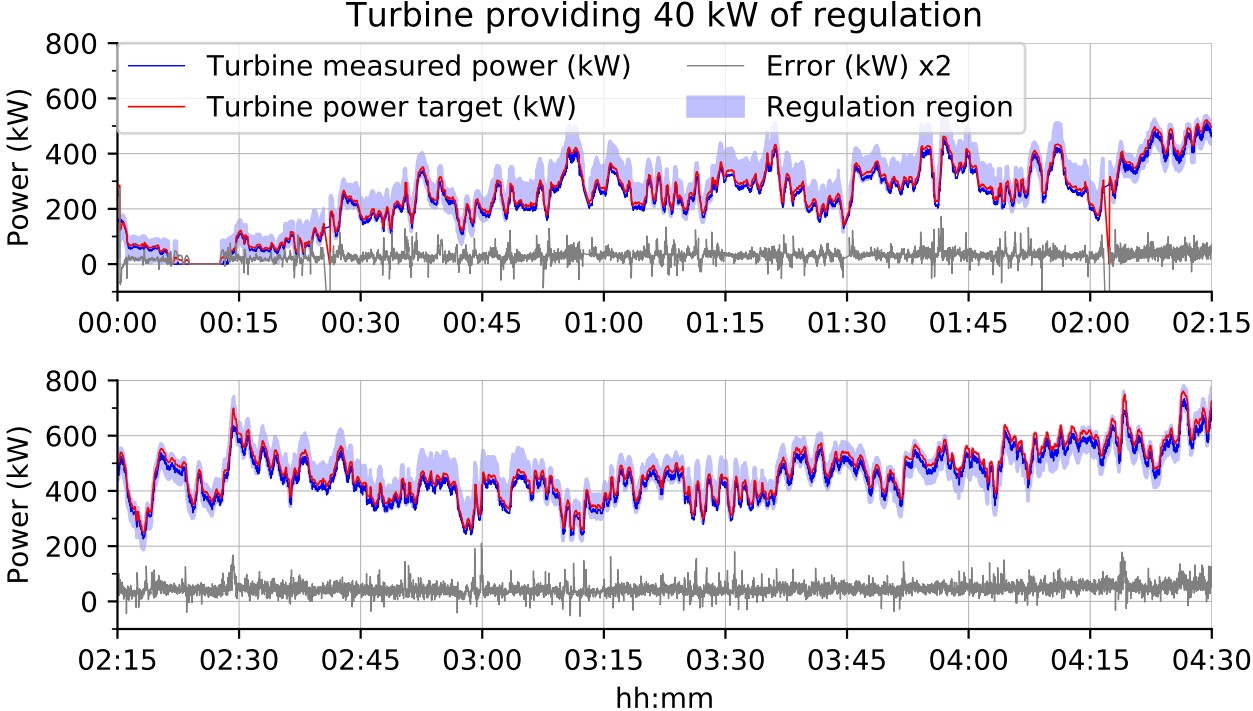

**Figure 9.** Test - 3: Providing 40 kW of regulation when wind speeds are below rated. Blue shaded region is the range of possible regulation. Note that the error value is multiplied by a factor of 2 for clarity.

of 40 kW is more significant than an error of 24 kW with a 100 kW regulation bid.

Additionally, note that using a wind turbine to provide AGC in the manner described in this work will produce some level of tracking error. The electrical power output of a Type 4 wind turbine is set by the power converter and its response time can be in the order of milliseconds. Power, via torque, is ultimately produced aerodynamically and is controlled by the wind turbine's pitch system, a system that has a typical response time of a few seconds. Depending on the control methods implemented, the pitch system also works to reduce the structural loads experienced by the turbine in addition to limiting power which may increase power tracking error. The question of wind turbine loads is an important aspect that requires further examination, particularly of field data during a demonstration such as presented here.

In addition to the effect of stochastic variations in turbine power output, our results were affected by a few other factors. One is time delays caused in part by the communication network and in part by the turbine's control system response time. Observations indicate that the net delay is approximately 12 seconds. A time delay negatively affects performance scores. Some of this is unavoidable as a wind turbine is a mechanical system with a finite response time. Another factor that affected the performance scores is data logging errors. The precise cause of these errors is not known but the

effect is that our logged data has several intervals where data is missing.

## 4 Financial Analysis

The previous sections examined the technical ability of a single Type 4 wind turbine in providing AGC. We now run a financial analysis to examine the profitability of providing secondary frequency regulation from a single wind turbine. We use PJM's ancillary services market for this to be consistent with our other works (Rebello et al. (2019) & Nasrolahpour et al. (2017)). Broadly, PJM's ancillary services market pays for providing capacity as well as payment proportional to performance (see Pilong (2015)). Capacity payments are for committing to provide ancillary services on the market and depend on the capacity (in MW) offered. Performance payments depend on the generator's performance when called upon to provide an ancillary service such as AGC. PJM's AGC market uses the metric of a performance score (0-100%) to calculate performance payments. The higher the performance score, the higher the payment received. PJM operate a competitive market for day-ahead energy and hour-ahead regulation. Generators bid into the market and are "cleared" depending on their price relative to other bidders. A good explanation of PJM's regulation market rules in the context of newer technologies can

be found in Xu et al. (2016). Additionally, PJM's AGC market provides two regulation signals for generators to follow: the faster moving Red-D signal and the slower Reg-A signal. The Reg-D signal is intended for technologies such as battery storage and sees more frequent changes to set-points. Mathematically, PJM refer to this movement as mileage. The AGC signal we use is simlar to PJM's Reg-D signal. For the calculations below, we assume that the wind turbine does not bid into the day-ahead energy market. Non-regulation energy is sold to the grid at the spot price. Regulation energy is sold at a separate price which depends on the amount of energy sold and the signal mileage ratio.

### 4.1    Inputs & assumptions

Inputs:

1. One year of 2017 PJM market data for spot prices and regulation (Available online at PJM's Data Miner website)

2. One year of historical power generation data from the Cowessess site. Note that this is power data, not wind speed data and therefore includes turbine down time

3. Performance scores (PJM) calculated earlier

Assumptions:

1. We use PJM's faster moving Reg-D signal as this is the regulation signal used for inverter-based technologies such as battery storage

2. We assume that the hourly average of the Reg-D signal is close to zero. This implies that the net effect of the regulation signal on average energy values is zero. Energy generated is affected solely by the curtailment applied (see Section 2.3). The effect of this curtailment is therefore to reduce energy payments by a constant value.

3. We use one year (2017) of historical power generation data from the Cowessess site. We then assume the hourly average power value to be a steady power generation value for that hour. The energy generated for that hour is therefore the average power value minus our curtailment. $(P_{avg}(t)\,kW - P_{curtailment}\,kW) \times 1h$

4. Regulation is provided each hour that the hourly average power is above the regulation offer (40 kW or 100 kW). From historical data, wind generation is greater than 40 kW for 74% of the year and is greater than 100 kW for 62% of the year.

5. We assume that the turbine is always cleared in the regulation market and that it is a price-taker i.e. it accepts prevailing market prices. This assumption leads to an upper limit on the possible regulation income.

6. We ignore maintenance costs as quantifying these effects is beyond the scope of this work

### 4.2    Results & discussion

The expected annual income with each of the performance scores calculated here is shown in Figure 7 (c). Observe that the total annual income when providing regulation is greater than the income from providing energy alone. This indicates that despite the limitations of this work, there is potential for even a single wind turbine to participate in the secondary frequency regulation market and for participation to be profitable.

The curtailment used represents energy which is not sold to the grid and therefore an opportunity cost to providing regulation services. Our calculations show that even with the lowest performance scores we calculated, regulation market income more than accounts for the lost energy cost. Any improvements to the performance score will only increase regulation market income.

Observe from Table 2 that the regulation market income with the improved performance score of Test 2* (100 kW) leads to a 6% increase in total income over supplying energy alone. The trade-off is lower income from energy sales due to the 100 kW curtailment required. This indicates that there is an incentive for even a single wind turbine to participate in the secondary frequency regulation market. Even with the lowest performance score obtained in Test 2 (59%), participating in PJM's secondary frequency market is still profitable. Although the additional income is modest, PJM regulation market prices account for the full opportunity cost of lost energy. With improvements in control algorithms and a reduction in turbine error, this will only be more favourable. This result is encouraging as even though wind generators are not required (or not allowed in some cases) to participate in the ancillary services markets, existing market structures make participation profitable. This situation may change when significant amounts of energy and ancillary services are supplied by renewable generators with a marginal fuel cost of zero.

## 5    Conclusions

This work presents the results of a series of tests evaluating the ability of a single, 800 kW Type 4 wind turbine to provide secondary frequency regulation (AGC). The turbine is located in Regina in the Canadian province of Saskatchewan. Tests are performed at and below rated wind speeds. The regulation offer is 10% of rated turbine power. We use a constant power curtailment to create room for up-regulation. Due to errors in the first series of tests, a second test was performed at rated wind speed however the regulation offer was changed to 25% of rated power. Performance scores of 59% and 65% are calculated with the PJM method, for wind speeds above and below rated respectively. Stochastic errors inherent in

**Table 2.** Annual financial summary with 2017 PJM market data (Also see Fig. 7)

| Test | $\eta_{PJM}$ (%) | Energy income ($) | Regulation capacity ($) | Regulation performance ($) | Total annual income ($) |
|---|---|---|---|---|---|
| Energy only | N/A | 61,464 | 0 | 0 | 61,464 |
| Test 2 - 40 kW | 59.4 | 59,331 | 1130 | 1400 | 61,861 |
| Test 2 - 100 kW | 83.9 | 56,131 | 3991 | 4942 | 65,064 |
| Test 3 | 65.4 | 59,331 | 1244 | 1541 | 62,116 |

the wind turbine's power output limit the performance scores achievable. With an increased regulation offer, we observe that the magnitude of stochastic error is relatively constant which leads to an improved performance score. Using the performance scores calculated, 2017 PJM market data and one year of historical site power generation data, we estimate the income possible from PJM's regulation market. We find that participating in the regulation market is profitable even with the lowest performance score, despite the opportunity cost of applying power curtailment.

**Data availability.** Data presented in this work was obtained through several Non-disclosure Agreements, however, results can be obtained by contacting the authors. Disclosing data sets may require signing NDAs.

**Competing interests.** We declare no competing interests

**Disclaimer.** The Wind Energy Institute of Canada makes no warranty, expressed or implied, and does not assume any legal responsibility for the accuracy of any information or processes described in this report. Reliance upon the information contained would be at the reader's sole and exclusive risk.

**Acknowledgements.** We would like to thank Natural Resources Canada's Office of Energy Research & Development for their financial support. We also thank the Alberta Electric System Operator for their time and data inputs. Additionally, we thank the Saskatchewan Research Council for allowing us to use their Cowessess First Nation's test site and also greatly appreciate their contributions of time and effort. Finally, we thank Enercon Canada for their support during this work and in refining our results.

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
