# Peer review of "Ancillary services from wind turbines: AGC from a single Type 4 turbine"

_Wind Energy Science, 2019_

## Referee Comment (RC1) · Malte Jansen (Referee) · 12 Jun 2019

The study presents an interesting outset, providing evidence of wind turbine capabilities. The style of writing is well understandable and clearly structured.

Literature: The study would have been better informed with a more detailed literature review. As this topic is on the intersect between academic research and industrial R&D, one might want to include more "grey" literature and conference proceedings, as the sole focus on academic publications is insufficient in identifying the current state of development. Without an expanded and comprehensive literature review, I cannot recommend the study for publishing. A more thorough literature review would deliver a large number of results for a large variety of countries and

regions, such as Germany, Ireland, Denmark and Texas. Some relevant pieces of literature to include: https://www.zhb-flensburg.de/dissert/jansen-malte/ (See page 63 for literature review table). https://ieeexplore.ieee.org/abstract/document/7521342 https://www.iee.fraunhofer.de/content/dam/iwes-neu/energiesystemtechnik/de/Dokumente/Studien-Reports/20140822_Abschlussbericht_rev1.pdf (in German but very relevant to the topic) https://ieeexplore.ieee.org/abstract/document/6038993 https://www.sciencedirect.com/science/article/pii/S1364032117309553?via%3Dihub https://scinapse.io/papers/2005811090 http://www.posspow.vindenergi.dtu.dk/

Methodology: In part the methodology is affected by the literature review and the study could have been designed better with the existing evidence in mind. The focus on the technical ability of wind turbines to deliver this kind of service add little novelty to the scientific community, as this already is part of the operational framework of many energy systems across the world. For example under www.regelleistung.net one can access the current conditions for wind to provide this service. Extensive field testing preceded these decisions of regulators and grid operators not just in Germany but across the world. The reader of the paper is led to believe that this is a new capability being tested on wind turbine, which is simply not true. The choice of a single 800 kW IEC Type 4 Turbine, which is presumably a E48 Enercon 800 kW low wind machine, appears as a non-representative choice. Given the variability of a single turbine and turbine type, the overall results are limited to a specific site and its conditions. It would be better to assess the capability of an entire wind farm, or preferably wind farm pool as this would better indicate the future value of wind providing reserves. Setting a 10% regulation band is not state of the art for offering this type of services. The low scores stem from this flaw and misrepresent the abilities of wind. Using probabilistic wind forecast is much more reliable way of determining the offering for wind. This leads to potentially wrong conclusions when comparing wind and fossil generators with regards to their reliability. Problems are being referred to, as e.g. wake effects and available active power, and being dismissed as out of scope. It would be good to have them at least referenced. Here the brevity of the literature review comes to show as the issues

have been researched before (e.g. PossPow project at DTU). It is unclear why AGC signal filtering is necessary and how it complies with PJM market rules. This makes it hard to understand the economic assessment. The economic evaluation could have benefitted from above mentioned PhD (Jansen) with it's open source modelling. A comparison with existing literature would have helped to understand the overall value, e.g. by how much does the income for the generator increase, what is the economic impact on the system, how big is the overall market? More general metrics would help to identify the magnitude of the technology. Disconnecting wind output from prices, i.e. using different areas for these, is a potentially difficult move. The prices of one price area should be correlated with the output of wind and solar. Thus using non-fitting time wind series to the price series would create a different baseline. At several stages, choices are not reasoned (e.g. why not use the P available of turbine?).

Writing: Whilst being comprehensive in most parts, the authors require to much previous knowledge from the reader. E.g. nowhere in the paper, it is explained what a IEC type 4 wind turbine looks like and why it is the right choice for the study. The prices and revenues for the PJM market drop seemingly out of nowhere and must be introduced to the reader, if the paper is to be read outside the US context. The paper seems to be taken out of a larger project. The author doesn't need to know about all the other tests that were conducted but not evaluated. Why start with "Test2" as the first one. That doesn't make sense. A critical reflection of the results is missing. This may be in part due to the aforementioned issues in the literature research. The reader needs to picked up a bit earlier. E.g. how is anyone supposed to know what the products in the PJM markets stand for without extensive research. These things are non-trivial.

---

## Author Comment (AC1) · 11 Jul 2019

Author's comments:

We thank the reviewer for their time and efforts in reviewing our paper and acknowledge their feedback.

We feel, however, that much of this review is based on a misunderstanding of the context and value of our work. The purpose of our work is not to document a new control algorithm or a new method of providing a grid service. Its goal is to add real-world, empirical data to existing work and to allow further studies to be more realistic instead of being based on assumptions and limited data. The value in this is that the turbine technology used is commercially available and is not experimental or a one-off

installation. Note that several other works use wind turbine performance scores derived from an NREL (National Renewable Energy Laboratory) test turbine i.e. one that is not commercially available.

We do not claim that the ability to provide AGC from wind generators is either new or novel. Instead, we claim that although the capability exists, it is either not utilised or under-utilised. Particularly when under-utilised, publicly available analyses are limited. Some of these are referenced in our paper. It is wrong to claim that '*The study would have been better informed with a more detailed literature* review*'* as the literature for our specific demonstration is extremely limited (see list below).

In other words, it is established that wind farms and wind turbines are able to vary their active power outputs in response to an external control signal. What is missing is an analysis of how well they are able to do this. What is the tracking error? What income can be obtained from providing such a service? What are the benefits to the power system? What are the long-term impacts on turbine components? Is a DFIG wind turbine better than a full-converter design? There are many unexplored issues and feel that it is premature to dismiss our work by claiming that *'this already is part of the operational framework of many energy systems across the world'.* The questions posed here are difficult to answer with simulations.

For context, we direct the reviewer to similar studies cited in our paper such as:

a. *Lew, Debra, et al. "Wind and solar curtailment: International experience and practices."* – See specifically Figure 4 and note that no analysis is provided. The figure is sourced from a presentation and is offered as-is.

b. *Loutan, Clyde, et al. Demonstration of essential reliability services by a 300-MW solar photovoltaic power plant.* – Although performed on a solar farm, this is very similar to our work and is relatively recent, proving our point about the lack of empirical data analysing ancillary services at a granular level
c. *Azpiri, Inigo, Clara Combarros, and Juan C. Perez. "Results from the wide-area voltage and secondary frequency control performed by wind power plants."* – This was a larger-scale demonstration of providing AGC from wind farms. Note that the analysis of the results is basic and that no financial analysis is performed.

Specific comments are addressed below:

1. This is not a study of primary frequency response or inertial response. Our analysis is specific to secondary frequency regulation, sometimes called AGC (automatic generation control).

2. We are aware of DTU's work in estimating the power available in the wind however, this is not the goal of this study. The goal of our work is to document and make public an analysis of the ability of a wind turbine to provide AGC. Our algorithm is simple by design.

3. We acknowledge the limitations of using a single wind turbine. We will, expand section 1.3 - Limitations to make the limitations more explicit.

4. Please be aware that organising a test campaign, especially one where power targets change every 4 seconds is not easy. The lack of published data is testament to this. There is, however, an abundance of simulation-based research work and simulations that draw on data from test turbines showing how even a single wind turbine can provide a service such as AGC. Data from commercial wind turbines is limited and our study is an attempt to change this.

5. *Setting a 10% regulation band is not state of the art for offering this type of services. The low scores stem from this flaw and misrepresent the abilities of wind* - The 10% regulation band was chosen to be consistent with existing work in this project and this is explicitly mentioned.

6. Note that despite some low scores, we demonstrate that much better performance is possible by using a larger regulation band (Test 2* - 200 kW regulation offer). We also demonstrate that tracking error remains relatively constant and that the percentage error is reduced with a larger regulation region. The reviewer's concern is already addressed in the paper. We disagree about this misrepresenting the capabilities of wind generators.

7. We disagree about the comparison with fossil-fuel generators (in terms of performance scores) being wrong. (See replies above)

8. Once again, we emphasise that real-world data is invaluable in future work and is difficult to obtain. Part of this barrier is the cost of infrastructure, the reluctance of owners to make their infrastructure available for research work and the reluctance of wind turbine OEMs to make such analysis data public.

9. Further, our previous work has already demonstrated providing AGC from a wind farm (as opposed to a single turbine) but please bear in mind that conducting such a demonstration even on a single wind turbine is already difficult and conducting it on a wind farm is harder still.

10. Much of the suggested literature:

    a. Deals with services other than AGC (mostly primary frequency response)
    b. Is simulation-based instead of using empirical data

    Please see the table at the end of this document.

11. The reviewer correctly claims that many grid operators have codes for wind generators to provide ancillary services. We do not dispute this claim. Even if extensive field testing preceded the introduction of these codes, much of the analysis is not public knowledge, specifically around secondary frequency regulation

(AGC). Note that we are referring specifically to ancillary services, not regulation reserves.

12. Several grid operators (or system operators):

    c. Have competitive ancillary service markets which technically allow the participation of wind generators but see close to zero participation from wind generators (PJM, for example). In other words, even if wind generators are allowed to participate, they do not.

    d. Have competitive ancillary service markets that exclude variable generators by design

    e. Simply mandate that wind generators be able to respond to an AGC signal when commanded. A good example of this is AEMO's (Australian Energy Market Operator) markets in Australia. A public analysis of how often such a feature is used and what its benefits are is often limited or non-existent.

    f. ENTSO-E's secondary control reserve regulations (linked here: https://www.entsoe.eu/fileadmin/user_upload/_library/publications/entsoe/ Operation_Handbook/Policy_1_final.pdf) do not specifically disqualify wind generators from participating the regulation market but where can find an actual analysis of their performance? How good or bad are wind generators are providing AGC, both up and down? Note that this is distinct from a negative control reserve which simply involves a relatively constant power curtailment. Providing AGC in a manner similar to fossil-fuel generators is a topic on which data and analysis is limited.

13. *Problems are being referred to, as e.g. wake effects and available active power, and being dismissed as out of scope. It would be good to have them at least referenced* – We acknowledge that these are issues are beyond the scope of our work and have already provided a starting reference to the topic (See section 1.3 – Limitations, for example).

14. *It is unclear why AGC signal filtering is necessary and how it complies with PJM market rules* – The specific need for AGC signal filtering is covered in section 2.1 – ACG signal filtering. We will clarify the implications of this filtering as related to PJM market rules in a future revision of the paper.

15. *At several stages, choices are not reasoned (e.g. why not use the P available of turbine?)* – We ill add reasons for this in a future revision of the paper. The reason behind this specific point is the error in the turbine's estimate of available active power.

16. *Why start with "Test2" as the first one. That doesn't make sense* – This is a trivial point. We specifically mention that we use the chosen test names to be consistent with other documentation from the same project.

17. *The prices and revenues for the PJM market drop seemingly out of nowhere and must be introduced to the reader, if the paper is to be read outside the US context.* – We will add more context to the PJM market mechanisms in a future revision.

    We question the validity of the suggested references as the majority appear to be unrelated to our work. A large proportion of the existing literature focuses on the issue of system inertia with high levels of wind generation and so deals with primary frequency response.

[]@ll@ https://www.zhb-flensburg.de/dissert/jansen-malte/ (See page 63 for literature review table). Relevant and was used in the preparation of this work, including the literature review table. Some of these references are cited in our works. The existing literature, however, has the exact drawback specified in our work i.e. it is largely simulation based and evaluates the broader-implications of providing control reserves / frequency regulation from wind generators. We agree with the broad claims but our work is more granular and is intended to serve as an input to studies such as those listed in this thesis.Economics of control reserve provision by fluctuating

renewable energy sources – Malte Jansen This is relevant to our work but is one step ahead i.e. it looks at the economics of providing control reserves from variable generators (wind, solar etc) on a competitive market. Only negative secondary control reserve is considered.https://www.iee.fraunhofer.de/content/dam/iwes-neu/energiesystemtechnik/de/Dokumente/Studien-Reports/20140822_Abschlussbericht_rev1.pdf (in German but very relevant to the topic) -System Inertial Frequency Response estimation and impact of renewable resources in ERCOT interconnection - Sandip Sharma ; Shun-Hsien Huang ; Ndr Sarma This focuses specifically on primary frequency response (aka fast-frequency response) from wind generators and is not relevant to our work.A review on frequency support provision by wind power plants: Current and future challenges - A.B.Attya, J.L.Dominguez-Garcia, O.Anaya-Lara Once again, this focuses on primary frequency response, not secondary frequency response. Further, this is entirely simulation-based and does not use empirical data.Kinetic energy and frequency response comparison for renewable generation systems Once again, this focuses on primary frequency response, not secondary frequency response.http://www.posspow.vindenergi.dtu.dk/ We are aware of this project but estimating the power available in the wind was not the focus of our work.

---

## Referee Comment (RC2) · Peiyuann Chen (Referee) · 31 Jul 2019

The paper evaluates mainly the use of Type 4 wind turbine to provide AGC by de-rated operation. This application has been discussed from time to time. In the reviewer's opinion, it is really not a good idea to ask the wind turbine to provide continuous AGC unless there is no other alternatives, which indicates a poor system design from the very beginning. First of all, it is important to investigate what the generation and storage fleets that are available in the system. Second, under what operational conditions, is it really necessary to use wind turbine to provide AGC by de-rating its operation? Why cannot hydro, biomass-based thermal or some forms of energy storage provide AGC instead? When can the wind turbine provide AGC and when can it not? What is the look-ahead time and how to deal with the forecast error? The reviewer believes these

are the critical questions to answer first. The technique of de-rating the wind turbine to provide AGC during wind conditions, may still need some improvement, is much clearer.

---

## Author Comment (AC2) · 7 Aug 2019

—- Author's comments We emphasise that although the question of providing various ancillary services from wind generators has been addressed before (albeit at a largely theoretical level), it has not been addressed at a granular level i.e. it has not been examined in the level of detail presented in our work. The broad questions of which other sources can provide AGC in a more cost-effective manner is not our focus. We focus solely on evaluating the ability of commercially available wind generation technology to provide AGC. Addressing topics such as when a wind turbine cannot provide AGC and forecast errors are beyond the scope of this work.

Specific comment: In the reviewer's opinion, it is really not a good idea to ask the

wind turbine to provide continuous AGC unless there is no other alternatives, which indicates a poor system design from the very beginning

The question of providing AGC from a wind turbine is very important to the design of a fossil-free grid and we feel that this does not indicate 'poor system design'. Whether or not a wind turbine is the best choice for providing regulation at any point is a separate question. —-

The question of de-rating a wind turbine to provide AGC is reasonable but this is not one that is directly relevant to our work. We focus primarily on the ability of the wind turbine to provide up- and down-regulation. How to improve this is a separate discussion entirely. This issue is already dealt with in the introduction to the paper.

The idea that other generation sources can provide AGC is valid but this does not invalidate the need to investigate the ability of wind generators to provide AGC.
* * *

---

## Referee Report (RR1)

Reviewer comments

The research idea and greatest addition of the paper is, as I interpret it, the benefit of a reduced and simplified technical part in combination with the financial aspects of providing secondary frequency control. This inorder to answer the question of how much money could be made, by a wind turbine owner operating its wind turbine in a derated operation, by offering ancillary services in the form of frequency control to the market.

Great work has been done to implement the PJM (but also NRC) performance score, the impact on the profitability could be clarified. The financial analysis leaves a lot more to be desired. More details are needed on the PJM ancillary services market (e.g. more information on how it works in practice, payments received, the ratio between capacity payments and performance payments).

The approach of combining the technical assessment and financial analysis is interesting but needs further work to represent a substantial contribution to scientific progress. Whilst it is the authors' aim to deliver high-level methods, this is too high-level to be of much value. Furthermore, the paper, as it is presented, would probably be of more value to wind-developers rather than system operators, which is the outlined intention of the authors.

The overall aim of the paper is not clearly stated, and seems to change focus from TSO/ISOs to owners of windfarms.

It would be interesting to see the contribution to: "[1]: Eldrich Rebello , David Watson, and Marianne Rodgers", Performance Analysis of a 10 MW Wind Farm in Providing Secondary Frequency Regulation: The technical difference between the derated operation from a type 3,4  or 5 WT, being pitch based, has no added difference or scientific value. The value of seeing a single turbine compared to the 5 presented in [1] is limited.

Similar work chapter, sentence 5-14 could be cut completely (due to lack of coherency/power variations/control options and as written erroneous, power electronics) or replaced with wind related frequency control papers. Or even an expansion of the similarities of [1].

The Limitations are well thought through, even though it would be interesting to observe the added ware on pitch actuators due to the proposed regulation. This since added maintenance would be a result of the suggested scheme.

AS a part of the motivation of the work, it is mentioned that :" *Our work is intended to make operational data public to allow for greater scrutiny by system / grid operators and to give grid operators an unbiased method of comparison between turbine technologies*". The interest should rather be on the owners of the turbines to get operational data, which they could get from wind turbine retailers.

CISO is given several abbreviations in the paper and should be "California Independent System Operator".

1.6. Test site and location, would benefit from removing the text regarding the battery. For coherency reasons, since it is not used in the paper at all there is no point in mentioning it (also shown in Figure 3), apart from possibly a last section on Future work in Conclusions.

2.1 AGC filtering section, is interesting but it would be interesting to see how the filter impacts the PJM/RRC-score. Furthermore, Reference article: [2] Rebello et. Al 2018, "Developing, implementing and testing up and down regulation to provide AGC from a 10 MW wind farm during varying wind

conditions" for the sizing of the 11 kW standard deviation is not solid. The standard deviation differs greatly depending on location (surface roughness), hub height and average wind speed.

Figure 3 would benefit from adding the AGC filter, updating frequencies of (AGC-signal and T-setpoint).

Table 1. Should contain Test 2*, the clarity of the section is not improved by removing it. That the test numbers are "kept consistent with other project documentation" is not that reasonable for a reader not associated with the other parts of the project.

Figure 6. Please separate the "regulation"-earnings into performance and capacity.

Regarding Figure 8. The error is always positive, and the regulation region seems to be often way above the power target. What is the main reason for this?

The paper appears to assume that delivering 80 kW of regulation in the PJM market means that 80 kW of capacity needs to be reserved (40 kW above and 40 kW below the operational midpoint). However, according to PJM Manual 12 Section 4.4, it appears that this would have been counted as 40 kW regulation rather than 80 kW. Would this imply that the revenues from regulation presented in the paper should be divided by 2?

The underlying reason for the financial calculation in the paper appears to be to compare the lost energy-market revenues due to curtailment to the revenues that would have been obtained from the PJM regulation market if the wind turbine sold regulation as often as feasible. However, since the PJM market co-optimizes energy and ancillary services, it should be possible to offer the wind turbine to the market in such a way that regulation bids would clear only during hours when it is more profitable than selling energy (assuming zero marginal cost for the incremental energy bid). Therefore, would it not have been more interesting to investigate how often regulation bids would have cleared and how much extra revenues this could generate?

From the PJM manuals it is unclear whether PJM would accept the methodology for calculating the performance score outlined in the paper. The default operational midpoint appears to be the 5-minute market setpoint. Under some circumstances the resource owner can send alternative operational midpoints to PJM (see Manual 12, 4.4.2). Would a wind turbine qualify for this?

Moreover, the assumptions on how and when regulation is or can be provided should be explained further. It would also be beneficial to add a discussion on the probability of regulation being provided at each hour that the hourly average power is above the regulation offer and how many hours per year that this is likely to occur.

---

## Author Response (AR2)

We wish to thank Mattias for his time and comments on our paper. All his comments were very useful, and he has made some important observations. Replies below are in red text. Changes to the paper are also in red text in the PDF.

**Reviewer** comments**

The research idea and greatest addition of the paper is, as I interpret it, the benefit of a reduced and simplified technical part in combination with the financial aspects of providing secondary frequency control. This inorder to answer the question of how much money could be made, by a wind turbine owner operating its wind turbine in a derated operation, by offering ancillary services in the form of frequency control to the market.

Great work has been done to implement the PJM (but also NRC) performance score, the impact on the profitability could be clarified. The financial analysis leaves a lot more to be desired. More details are needed on the PJM ancillary services market (e.g. more information on how it works in practice, payments received, the ratio between capacity payments and performance payments).

**The section about the PJM ancillary services market has been expanded. We also included a reference which examines the PJM regulation market in the context of new generation technologies. We have not repeated much of the referenced text for the sake of brevity.**

The approach of combining the technical assessment and financial analysis is interesting but needs further work to represent a substantial contribution to scientific progress. Whilst it is the authors' aim to deliver high-level methods, this is too high-level to be of much value. Furthermore, the paper, as it is presented, would probably be of more value to wind-developers rather than system operators, which is the outlined intention of the authors.

This is incorrect but we have included additional context where relevant (Sections 1.2, 1.3 & 1.4). The primary focus of the paper is system operators. The primary aim of the paper is to make public a granular analysis of wind turbine response when providing AGC. This is currently missing in the published literature and there exist publications based on performance scores calculated from an NREL simulation i.e. not field data. Our aim is to make performance data available for future research so that it can be based on field measurements as opposed to simulations.

The overall aim of the paper is not clearly stated, and seems to change focus from TSO/ISOs to owners of windfarms.

**See reply above. This is now corrected.**

It would be interesting to see the contribution to: "[1]: Eldrich Rebello , David Watson, and Marianne Rodgers", Performance Analysis of a 10 MW Wind Farm in Providing Secondary Frequency Regulation: The technical difference between the derated operation from a type 3,4 or 5 WT, being pitch based, has no added difference or scientific value. The value of seeing a single turbine compared to the 5 presented in [1] is limited.

We disagree. Although the method of rotor speed control between IEC turbine types is often through blade pitch, there is significant difference in the electrical power delivery method and therefore the measured electrical response. The revised manuscript expands on this point and includes diagrams of IEC Type 3 and 4 wind turbines.

Similar work chapter, sentence 5-14 could be cut completely (due to lack of coherency/power variations/control options and as written erroneous, power electronics) or replaced with wind related frequency control papers. Or even an expansion of the similarities of [1].

We have expanded Section 1.2 to be clearer. Some of these changes were in response to comments from other reviewers.

The Limitations are well thought through, even though it would be interesting to observe the added ware on pitch actuators due to the proposed regulation. This since added maintenance would be a result of the suggested scheme.

We agree however more data is required to comment on this. A more significant problem is that the internal data logging interval on the wind turbine used was 30s. For an accurate measure of pitch travel (in degrees), 1s or better sampling is required.

AS a part of the motivation of the work, it is mentioned that :" *Our work is intended to make operational data public to allow for greater scrutiny by system / grid operators and to give grid operators an unbiased method of comparison between turbine technologies*". The interest should rather be on the owners of the turbines to get operational data, which they could get from wind turbine retailers.

We disagree. As mentioned in the revised manuscript (Section 1.2), there is an inherent conflict of interest in wind turbine OEMs publishing these numbers. Further, turbine OEMs simply comply with system operator rules and are unlikely to participate in an unbiased performance review (as proven by the lack of public data).

CISO is given several abbreviations in the paper and should be "California Independent System Operator".

**Corrected.**

1.6. Test site and location, would benefit from removing the text regarding the battery. For coherency reasons, since it is not used in the paper at all there is no point in mentioning it (also shown in Figure 3), apart from possibly a last section on Future work in Conclusions.

**Revised text. Figure 3 (now Figure 4) also revised.**

2.1 AGC filtering section, is interesting but it would be interesting to see how the filter impacts the PJM/RRC-score. Furthermore, Reference article: [2] Rebello et. Al 2018, "Developing, implementing and testing up and down regulation to provide AGC from a 10 MW wind farm during varying wind conditions" for the sizing of the 11 kW standard deviation is not solid. The standard deviation differs greatly depending on location (surface roughness), hub height and average wind speed.

Text revised (See section 2.1). We agree that the standard deviation value calculated is valid only for this turbine in this particular site.

Figure 3 would benefit from adding the AGC filter, updating frequencies of (AGC-signal and T-setpoint).

**Revised to include update frequencies.**

Table 1. Should contain Test 2\*, the clarity of the section is not improved by removing it. That the test numbers are "kept consistent with other project documentation" is not that reasonable for a reader not associated with the other parts of the project.

**Revised.**

Figure 6. Please separate the "regulation"-earnings into performance and capacity.

**Revised.**

Regarding Figure 8. The error is always positive, and the regulation region seems to be often way above the power target. What is the main reason for this?

This was an oversight in the figure legend. The error shown is the absolute value of error between the target power and the measured power. The figure is difficult to read when the actual error value is plotted (values are small and therefore misleading).

The nature of the regulation region is due to the nature of the AGC signal used. Note from Figure 6 (b) that the regulation signal is negative more often than it is positive.

The paper appears to assume that delivering 80 kW of regulation in the PJM market means that 80 kW of capacity needs to be reserved (40 kW above and 40 kW below the operational midpoint). However, according to PJM Manual 12 Section 4.4, it appears that this would have been counted as 40 kW regulation rather than 80 kW. Would this imply that the revenues from regulation presented in the paper should be divided by 2?

You are correct. The source of our confusion was Exhibit 14 on page 51 of PJM manual 12. The revenue numbers are now revised but the reduction is less than 50%. This is due to slightly increased income from the energy market.

The underlying reason for the financial calculation in the paper appears to be to compare the lost energy-market revenues due to curtailment to the revenues that would have been obtained from the PJM regulation market if the wind turbine sold regulation as often as feasible. However, since the PJM market co-optimizes energy and ancillary services, it should be possible to offer the wind turbine to the market in such a way that regulation bids would clear only during hours when it is more profitable than selling energy (assuming zero marginal cost for the incremental energy bid). Therefore, would it not have been more interesting to investigate how often regulation bids would have cleared and how much extra revenues this could generate?

This is a valid question but is unfortunately beyond the scope of this paper. We choose to focus primarily on making time-series performance data public. The regulation market income calculated in this paper represents and upper limit on the possible income from the regulation market. This follows from assumption #5 in Section 4.1 i.e. the turbine's regulation market bid is always accepted when it is able to provide regulating power.

From the PJM manuals it is unclear whether PJM would accept the methodology for calculating the performance score outlined in the paper. The default operational midpoint appears to be the 5-minute market setpoint. Under some circumstances the resource owner can send alternative operational midpoints to PJM (see Manual 12, 4.4.2). Would a wind turbine qualify for this?

This is possible but is uncommon. At the moment, PJM generators change their operational midpoint a few times an hour, at most. Part of the reason is that wind generators do not currently provide AGC on the PJM market.

Moreover, the assumptions on how and when regulation is or can be provided should be explained further. It would also be beneficial to add a discussion on the probability of regulation being provided at each hour that the hourly average power is above the regulation offer and how many hours per year that this is likely to occur.

We have expanded on the assumptions in Section 4.1 to be clearer. We have also added numbers (%) to show how often the wind turbine is able to provide regulation (Point #4).

---

## Author Response (AR3)

Replies to editor comments, December 5, 2019.

The authors wish to thank the editor for the comments below. Each comment is addressed separately and the relevant text from the paper is copied where relevant to make review easier. Changes to the manuscripts are indicated in red text where applicable.

Please note that the submitted PDF may have figures placed at the end of the document. This is due to Latex restrictions on figure placement in a single column document and can be corrected in the final print version (2 column).

Dear Authors,
I like your work with testes on commercial wind turbines it gives a good knowledge and experience of the control of the turbine and can help the development of the frequency control by wind turbines.
I have seen your last update of the paper and it is good, although I still like to see some small updates.

**1. Regarding the comments from the latest review. Figure 2. Have a figure of Type 3 and Type 4 wind turbine and saying they ae different, it give notthing to the reader. Skip this text.**

*Updated*.

**2. I like to see a motivation to your selection of AGC signals in figure 6. One is at rated power the other one around zero, wat what is good with the selection, what should the reader learn.**

*Added. To make review simpler, the text is copied below:*

*The Alberta Electric System Operator (AESO) provided both AGC signals used in this work. One was a 30 minute duration signal and the other was 4.5 hours long. These are identical to the signals used in Rebello et al. (2019) and Nasrolahpour et al. (2017). This is done to make direct comparison with earlier work easier. Both signals use a 4 s update interval which is 5 identical to PJM's Reg-D signal. Although we do not use PJM's regulation signals, the identical update intervals allows for a more straightforward comparison. The first step to signal preparation was scaling the raw AGC signals to fit within our chosen regulation ranges. The results of this scaling are shown by blue traces in Figure 5 (a) and (b). Note that the signal in Figure 5 (a) has a range from 720 to 800 kW (centred around 760 kW i.e. 800 ⊠ 40 kW) as these power targets are sent directly to the wind turbine. Power values in the range [720, 800] kW are within the operational range of the wind turbine and this test is 10 performed when prevailing wind speeds are above the turbine's rated wind speed i.e. rates power production is possible. The signal in Figure 5 (b) is centred around zero kW as this signal is a bias value. The bias values therefore in the range of [-40,40] kW and are added to an estimated power value as described in Section 2.3. The scaling process was followed by filtering, as described below. 3.*

**3. In the same way what are the motivations for perform the specific tests I table 1 ?**

*Added. To make review simpler, the text is copied below:*

*The experiments presented in this work are grouped into two tests as summarized in Table 1 with two being above rated wind speed and one below. The aim of both tests is to examine the ability of the wind turbine to vary its active power output in 20 response to an external target. In order to provide a complete picture, examining this ability across the full range of operational wind speeds is required. Test 3 is*

*performed below rated wind speed and therefore requires a varying power curtailment to provide up-regulation. As described in Section 2.3, this varying power curtailment is provided via a wind speed estimate and a power curve. In contrast, Test 2 is performed when prevailing wind speeds are above the turbine's rated wind speed and rated power production is therefore possible. No estimate of power production is required. We also present a variation of Test 2 where the regulation offer is 100 kW. This is denoted by Test 2\* as it is functionally identical to Test 2, the only difference being a larger regulation region (100 kW versus 40 kW in Test 2). Test numbers are kept consistent with other project documentation. The two tests presented here are the only ones with the wind turbine operating independently.*

**4. Page 9 line 3: some words missing?**

Corrected

**5. Page 9 line 5: Do not write about a future battery test in this paper, may be in a specific chapter: Future work**

Removed mention of the battery. It is still included in the system description (Section 1.6) for completeness

**6. Page 12 Figure 9: Nice if you could add an error signal in figure 9, as you have done in figure 10.**

Done.

**7. Page 13: Figure 10. Please improve the color for the regulation region, very hard to read. I also suggest that you have a separate scale for the Error, on the right side of the figure, maybe with a factor 4 to be able to read the Error. Alternative have a special figure for the Error. The Error is an important number and need to be readable.**

Done**.**

**And then a comment to the discussion of electrical torque/power: The electrical power output is set by the control system of the converter of the wind turbine and it can be very fast down to some ms. The pitch controller is used to limit the speed of the turbine by reducing the power input to the turbine. And there by also reducing the incoming loads of the turbine.**

Included in Section 3.5 and copied below for reference:

*Additionally, note that using a wind turbine to provide AGC in the manner described in this work will produce some level of tracking error. The electrical power output of a Type 4 wind turbine is set by the power converter and its response time can be in the order of milliseconds. Power, via torque, is ultimately produced aerodynamically and is controlled by the wind turbine's pitch system, a system that has a typical response time of a few seconds. Depending on the control methods implemented, the pitch system also works to reduce the loads experienced by the turbine in addition to limiting power which may increase*

*power tracking error. The question of wind turbine loads is an important aspect that requires further examination, particularly of field data during a demonstration such as presented here.*